# STKOpt: Automated Spatio-Temporal Knowledge Optimization for Traffic Prediction

## Abstract

Ubiquitous sensors and mobile devices have spurred the growth of Web-of-Things (WoT) services in smart cities, making accurate spatio-temporal traffic predictions increasingly crucial. Leveraging advances in deep learning, recent Spatio-Temporal Graph Neural Networks (STGNNs) have achieved remarkable results. However, these methods address scenario-specific spatio-temporal heterogeneity by designing model architectures, often overlooking the importance of selecting optimal spatio-temporal knowledge (i.e., model inputs). In this paper, we propose an automated framework for spatio-temporal knowledge optimization to address this challenge. Our framework seamlessly integrates with downstream models, enhancing their performance across various prediction tasks. Specifically, we design a knowledge search space composed of parameters that represent scenario-specific spatio-temporal correlations within data. Additionally, we employ a bandit-based multi-fidelity algorithm for knowledge optimization to solve the constraint of limited resource. Furthermore, we adopt a meta-learner to extract transferable meta-knowledge about optimal knowledge, facilitating efficient exploration of the search space. Extensive experiments on five widely used real-world datasets demonstrate the effectiveness of our proposed framework. To the best of our knowledge, we are the first to automatically optimize spatio-temporal knowledge for spatio-temporal traffic prediction.

## CCS Concepts

• **Computing methodologies → Machine learning**; • **Information systems → Spatial-temporal systems**.

## Keywords

Traffic Prediction, Spatio-Temporal Modeling, Automated Machine Learning

### ACM Reference Format:

Anonymous Author(s). 2024. STKOpt: Automated Spatio-Temporal Knowledge Optimization for Traffic Prediction. In . ACM, New York, NY, USA, 12 pages. https://doi.org/10.1145/nnnnnnn.nnnnnnn

## 1 Introduction

As sensors, mobile devices, and positioning technologies become increasingly prevalent, a surge of innovative Web-of-Things (WoT) services, such as shared bicycle system and ride-hailing platforms,

have become integral components of smart cities. Since these services heavily rely on spatial and temporal human mobility dynamics, accurate spatio-temporal traffic prediction plays a pivotal role in intelligently monitoring traffic systems [22, 29, 30, 45] and offering crucial insights for traffic control [18], emergency response [45] and urban planning [41].

The challenge of spatio-temporal traffic prediction lies in capturing scenario-specific ST heterogenity. Recently, the Spatio-Temporal Graph Neural Networks (STGNNs) have garnered significant attention, whose basic framework for predictive learning mainly contains data processing and spatio-temporal graph (STG) learning module [11]. Since data from diverse scenarios shows heterogeneous ST correlations, it can be theoretically addressed from two key aspects. Firstly, in the data processing module, constructing the STG data that best captures the underlying data distribution as model inputs, including historical sequences and corresponding graph structures that we called ST knowledge. Secondly, in the STG learning module, designing the optimal network architecture for ST dependency modeling.

To address the issue of scenario-specific ST heterogeneity, existing methods have achieved considerable advancements by designing STG learning modules, i.e., model architectures. For example, attention mechanism have been adopted for modeling scenario-specific temporal correlation [12, 39]; adaptive graph learning methods, including learnable static graphs [1, 35] and time-varying dynamic graphs [8, 20], have been proposed to learn application-specific spatial correlation. The essence of these methods lies in optimizing model parameters. Furthermore, neural architecture search (NAS) methods have been introduced to automatically design optimal model architecture for specific datasets [15, 23, 33]. However, these methods overlook the importance of selecting optimal model inputs (i.e., ST knowledge), leading to the following practical issues. **i) Lack of scenario-specific customized temporal knowledge.** There are multiple types of temporal correlations among different time intervals [8, 28]. Taking traffic flow as an example, the impact duration of congestion varies between scenarios such as first-tier cities and smaller cities. Most existing studies rely solely on the most recent time steps (e.g., past 12 time steps [13, 26]). However, it has been proven that incorporating effective temporal knowledge would significantly improve prediction [8, 19, 32]. **ii) Lack of scenario-specific customized spatial knowledge.** Constructing spatial topology is crucial for capturing spatial heterogeneity. Predefined graphs can be constructed using multiple functions such as distance [40], connectivity [6], and points of interest (POI) similarity [6]. However, the specific data conditions vary across different scenarios, for predefined graph-based methods, it is difficult to manually select well-work predefined graphs for different datasets. In addition, for adaptive graph-based methods, they can benefit further from predefined graphs [13, 37].

Another idea to address the issue of scenario-specific ST heterogeneity is to optimize ST knowledge, i.e., selecting the optimal ST

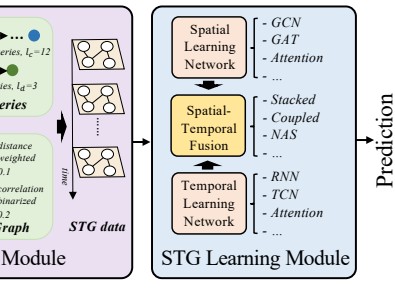

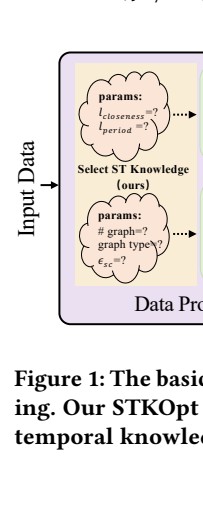

**Figure 1: The basic framework of STGNN for predictive learning. Our STKOpt focuses on how to obtain optimal spatio-temporal knowledge in the data processing module.**

knowledge for different scenarios. However, currently researchers can still only manually select knowledge, which demands substantial domain knowledge and human efforts. Therefore, there is a strong motivation to design an automated ST knowledge optimization framework for various application scenarios. Fortunately, recent automated machine learning (AutoML) technology has been widely applied across various fields [2, 31], including the field of ST traffic prediction [15, 23, 33]. However, adopting existing automated methods to optimize ST knowledge presents two main challenges. Firstly, optimal ST knowledge must accurately represent the ST correlations within data. Therefore, it is crucial for the search space to be as extensive as possible, covering the possibilities of various data distributions. Existing NAS methods considered the structure of the adjacency matrix as a special architecture in the proposed model, employing one-shot methods simultaneously evaluate all candidate matrices [10, 14]. However, this approach places significant demands on computational resources, thereby constraining the scalability of the search space. Secondly, given the high dimensionality of the knowledge search space, finding optimal configurations necessitates a substantial number of evaluations. The complexity of ST traffic prediction models further exacerbates this issue, as random search and Bayesian methods demand extensive computational resources to effectively explore the search space. This requirement poses significant challenges for researchers with limited budgets. To tackle aforementioned challenges, we propose a novel framework for automated ST knowledge optimization, entitled STKOpt. Notably, STKOpt is a general framework that can seamlessly integrate into existing STGNNs. Our main contributions are summarized as follows:

- We propose a general framework for spatio-temporal traffic prediction, entitled STKOpt, focusing on automated ST knowledge optimization. This framework seamlessly integrates into existing STGNNs, enhancing their performance. To the best of our knowledge, we are the first to automatically discover optimal ST knowledge for spatio-temporal traffic prediction.
- To accommodate various scenario-specific data distributions, we atomize ST knowledge and design a knowledge search space composed of parameters. To solve the constraint of limited resource, we employ a multi-fidelity optimization strategy that

leverages downsampling to increase the number of configurations evaluated. To efficiently explore search space, a meta-learner is adopted to extract transferable meta-knowledge about optimal ST knowledge.
- We conduct extensive experiments on real-world datasets, and the experimental results demonstrate that our STKOpt can enhance the performance of downstream STGNNs. In particular, the improved prediction we achieve in this paper will enhance the confidence and proactiveness of users for emerging WoT services. The knowledge optimization ideas we propose can be broadly applicable for web-based ubiquitous computing, including environment analysis and energy management, as well as contributing to the development of smart cities [44].

## 2 Related Work

### 2.1 Deep Learning for Spatio-Temporal Traffic Prediction.

In recent explorations of spatio-temporal traffic prediction, deep learning technology has become a mainstream trend. Almost all studies consider the effect of recent time intervals, and some studies further consider multiple periodicities [8, 28, 32, 42], demonstrating its effectiveness. Due to the importance of prior knowledge, early studies introduced predefined graphs, including distance-based weighted graphs[17, 28] and binary graph constructed based on connectivity of nodes [6, 7, 27]. In order to make up for the lack of prior knowledge, subsequent studies introduced learnable graph structures to model implicit spatial correlations [1, 36, 43]. Recently, attention mechanism has been widely adopted to model dynamic graphs [7–9]. However, most of them still require predefined graph structure as input [9, 20, 24, 43]. In addition, some models try to combine predefined graphs and adaptive graphs, which can better combine the advantages of both [13, 37]. For different scenarios, there are still research gaps on which temporal factors to consider and how to find the most effective predefined graphs to improve the accuracy of downstream prediction tasks. Our STKOpt aims to address this limitation.

### 2.2 Automated Machine Learning for Spatio-Temporal Traffic Prediction.

Recently, automated machine learning (AutoML) has been extensively explored in the field of spatio-temporal (ST) traffic prediction. Most studies aim to automatically find the optimal ST modeling architecture for specific prediction tasks, utilizing neural architecture search (NAS) methods [15, 23, 33]. Some studies also focus on automatically finding the optimal graph structure [10, 14]. These works treat the adjacency matrix of the predefined graph as a special architecture of the model and use one-shot NAS methods to evaluate all candidate matrices at once, identifying the best one. This approach requires memory proportional to the number of candidate matrices, limiting the expansion of the search space. Additionally, some studies emphasize the selection of optimal hyperparameters, finding them through joint search with the architecture [34] or using ranking strategies [38]. However, these approaches cannot be directly applied to optimizing ST knowledge because we require different

search spaces to construct time-series and graph structures that accurately reflect the data distribution as model input.

## 3 Preliminaries

### 3.1 Definitions and Problem Statement

*Definition 3.1.* **Spatio-Temporal Graph.** A spatio-temporal graph (STG), denoted as $\mathcal{G} = (\mathcal{V}, \mathcal{E}, \mathbf{A})$, where $\mathcal{V}$ is a set of $N = |\mathcal{V}|$ nodes that represent different locations and have time-varying attributes; $\mathcal{E}$ is a set of edges and $\mathbf{A} \in \mathbb{A}^{N \times N}$ is a adjacency matrix where each element represents the inter-location relationships between nodes (e.g., distance, connectivity). The attributes over an STG can be viewed as a graph signal $\mathbf{X} \in \mathbb{R}^{T \times N \times C}$, where $T$ is the length of sequence, $C$ is the number of attributes of each node. The graph signal is $\mathbf{X}_t \in \mathbb{R}^{N \times C}$ at each time step $t$.

*Definition 3.2.* **Spatio-Temporal Knowledge.** ST knowledge is constructed from raw data in the data processing module and serves as input to the subsequent STG learning network. Specifically, *temporal knowledge* is represented as a collection of time-series data sampled at various intervals, capturing distinct temporal dynamics: $\mathbf{T} = \{\mathbf{X}_{p_k} | p_k \in P\}$. Here, $P = \{p_1, p_2, ..., p_n\}$ represents a set of specific intervals, each $p_k$ denoting a particular sampling period. The time-series data for each period $\mathbf{X}_{p_k}$ is defined as:

$$\mathbf{X}_{p_k} = [\mathbf{X}_{t - i \cdot p_k} | i = 1, \dots, l_k], \tag{1}$$

where $l_k$ denotes the length of the time series associated with the period $p_k$, and $t$ is the current time step; *spatial knowledge* is composed of a series of predefined graph, denoted as $\mathbb{M}_{\mathcal{G}} = \{\mathcal{G}_i | i = 1, \cdots, k\}$, where $k$ is the number of selected graphs.

*Definition 3.3.* **Spatio-Temporal Knowledge Optimization.** Our STKOpt aims to improve the accuracy of downstream prediction tasks by finding optimal ST knowledge, which achieves the minimum loss on validation dataset $\mathbb{D}_{val}$:

$$\min_{\lambda^*} \mathcal{L}(\lambda^*, \mathbb{D}_{val}) \quad \text{s.t.} \quad \lambda^* = \arg \min_{\lambda \in \Lambda} \mathcal{L}(\lambda, \mathbb{D}_{val}) \tag{2}$$

where $\mathcal{L}$ denotes loss function, $\lambda$ denotes a specific configuration within the set $\Lambda$ of all possible ST configurations, $\lambda^*$ is the optimal configuration that minimizes the loss.

### 3.2 Multi-fidelity Optimization

Multi-fidelity optimization effectively addresses limited time and resource constraints by combining evaluations of varying fidelities. Lower-fidelity evaluations are less costly but offer poorer generalization, while higher-fidelity evaluations, though more expensive, yield better generalization. one representative bandit-based approach is HYPERBAND [16], which forms the basis of our proposed optimization framework. SUCCESSIVE HALVING, as a sub-routine within HYPERBAND, uniformly allocates a budget ($B$) to a set of configurations ($n$), evaluates their performance, discards the worst half, and repeats until only one configuration remains. HYPERBAND explores multiple values of $n$ for a given $B$ by partitioning the budget into combinations of $n$ and minimum resource $r$, and then executing SUCCESSIVE HALVING for each combination

($n_s, r_s$). In the next section, we improve the design of the configuration sampling method to more effectively identify promising configurations within the extensive search space.

## 4 Methodology

In this section, we introduce the details of the STKOpt, which accepts three hyperparameters as input: (1) $b$, one unit of resource, i.e., the minimum desired resource; (2) $B$, the ratio between maximum resource and minimum resource; (3) $\eta$, the proportion of configurations discarded in each round. Following HYPERBAND, the workflow of STKOpt consists of a loop that iterates over different configuration numbers ($n$) and minimum resources ($r$). For each sub-loop ($s$) with a fixed ($n_s, r_s$), the framework can be divided into a sampling stage and an evaluation stage, as shown in Figure 2.

**Sampling stage.** The *knowledge generator* employs random strategy or a meta-learner to sample $n_s$ ST knowledge configurations from the *knowledge search space*.

**Evaluation stage.** The *knowledge evaluator* uses the SUCCESSIVE HALVING algorithm to evaluate the received $n_s$ configurations based on the budget $r_s$ to find the best configuration.

Algorithm 1 illustrates the complete spatio-temporal knowledge optimization process. The input $B$ determines how many sub-loops are considered. Specifically, $s_{max} + 1$ different values for $n$ are considered with $s_{max} = \lfloor \log_\eta(B) \rfloor$.

---

**Algorithm 1** Knowledge Optimization

---

**Require:** maximum budget $B$, one unit of budget $b$ and $\eta$
**Ensure:** optimal knowledge configuration
1: $s_{max} = \lfloor \log_\eta(B) \rfloor$
2: **for** $s \in \{s_{max}, s_{max} - 1, \dots, 0\}$ **do**
3:      $n_s = \lceil \eta^s \frac{s_{max}+1}{s+1} \rceil, \quad r_s = B\eta^{-s}b$
4:      Sample $|\Lambda| = n_s$ configurations
5:      **for** $i \in \{0, \dots, s\}$ **do**
6:          $n_{s,i} = \lfloor n_s \eta^{-i} \rfloor, r_{s,i} = r_s \eta^i$
7:          $loss = \{\mathcal{L}(\mathcal{M}(\lambda), r_{s,i}, \mathbb{D}_{val}) : \lambda \in \Lambda\}$
8:          $\Lambda = topk(\Lambda, loss, \lfloor n_{s,i}/\eta \rfloor)$
9:      **end for**
10: **end for**
11: **return** Configuration with lowest loss so far.

---

### 4.1 Knowledge Search Space Design

To accommodate diverse scenario-specific data distributions, we atomize ST knowledge into individually adjustable and evaluable parameters. This approach enables researchers to finely control and optimize both the composition of time series data and the structure of graphs, thereby tailoring them to meet the precise requirements of specific prediction tasks. Based on this atomization, we propose a knowledge search space composed of these parameters, as shown in Table 1.

*4.1.1 Temporal Knowledge.* In traffic prediction, temporal knowledge can be summarized into three categories: closeness, daily periodicity and weekly periodicity due to their effectiveness in capturing temporal correlations among different time intervals [28, 32, 42]. Each category is designed to capture specific temporal dynamics. (1)

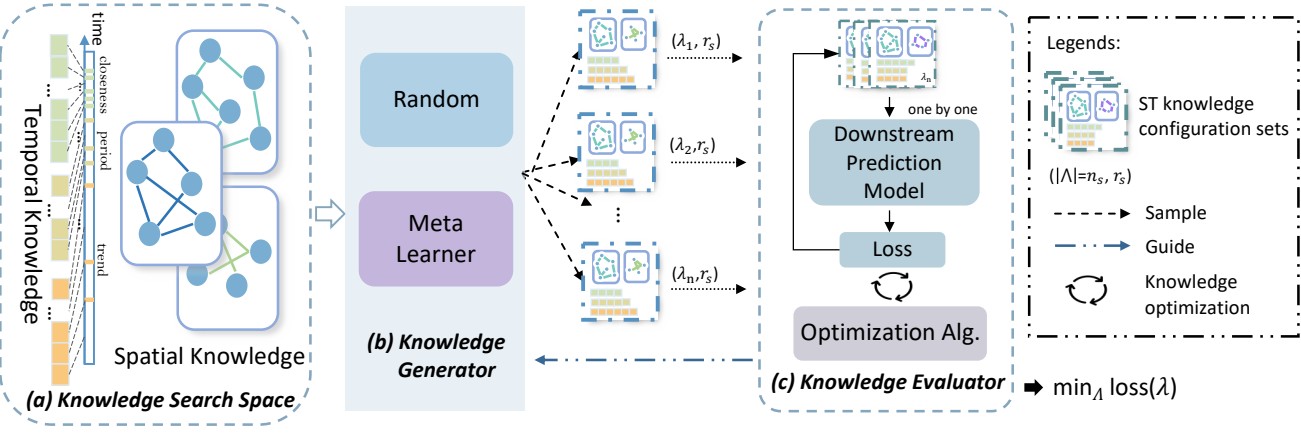

Figure 2: Workflow of each sub-loop in STKOpt.

Table 1: Knowledge Search Space. (#$graph$: number of graphs; $\mathcal{G}_{type(i)}$: type of adjacency matrix for $\mathcal{G}_i$; $\epsilon_{sc(i)}$: sparsity coefficient for $\mathcal{G}_i$)

| Knowledge | Values |
|---|---|
| $l_c$ | $1, \cdots, 12$ |
| $l_d$ | $0, \cdots, 7$ |
| $l_w$ | $0, \cdots, 4$ |
| # graph | $1, \cdots, n$ |
| $\mathcal{G}_{type(i)}$ | binarized,weighted |
| $\epsilon_{sc(i)}$ | 0.1,0.2,0.3,0.4 |

closeness: $\mathbf{X}_c = [\mathbf{X}_{t-l_c}, \mathbf{X}_{t-(l_c-1)}, \ldots, \mathbf{X}_{t-1}]$, captures short-term dependencies that are immediately adjacent to the predicting period. (2) daily periodicity: $\mathbf{X}_d = [\mathbf{X}_{t-l_d \cdot d}, \mathbf{X}_{t-(l_d-1)\cdot d}, \ldots, \mathbf{X}_{t-d}]$, captures the daily repeated patterns. (3) weekly periodicity: $\mathbf{X}_w = [\mathbf{X}_{t-l_w \cdot w}, \mathbf{X}_{t-(l_w-1)\cdot w}, \ldots, \mathbf{X}_{t-w}]$, captures the weekly periodic features. Here, $d$ and $w$ are period and trend spans that are fixed to one-day and one-week respectively. $l_c, l_d$ and $l_w$ are input lengths of the closeness, daily periodicity and weekly periodicity time-series, respectively.

We note in many existing works, the length of the closeness typically does not go beyond 12, as longer temporal dependencies are generally not observed in the most recent time slots. We limit the daily periodicity length to 7 as it encompasses the traffic patterns for all seven days of the week. Similarly, we cap the length of weekly periodicity at 4, considering that monthly patterns are generally less pronounced. Furthermore, in certain application scenarios where the data lacks periodicity, we allow the corresponding periodicity sequence length to be set to zero.

4.1.2 *Spatial Knowledge.* Despite different inter-location relationship graphs can represent many types of spatial relationships, existing work [32] has proven that blindly adding various spatial knowledge graphs to the model may not help network parameter learning,

and even reduce prediction performance and computational efficiency. Therefore, determining the number of spatial knowledge graphs and which ones to employ are the two important parameters that need to be optimized.

Upon conducting a comprehensive review of existing literature, we observe that the construction method of adjacency matrices generally falls into two predominant categories: threshold-based binarization and direct weight assignment. Formally, The relationship between location $i$ and $j$ is denoted as $w_{i,j} = \text{relationship}(i, j)$. Hence, we can build binarized graphs by Equation (3) and weighted graphs by Equation (4).

For binarized graphs, the simple graph structure endows it with higher computational efficiency and reduced sensitivity to data noise. Furthermore, by adjusting the sparsity coefficient, the graph structure can more accurately match the characteristics of specific data distributions. In contrast, weighted graphs can capture multi-dimensional spatial correlations with their rich weight information. The choice between these structures largely hinges on the data distribution. Sparse binarized graphs excel in scenarios with robust local correlations, capturing essential relationships and filtering out potential noise introduced by weighted graphs. On the other hand, in cases with a more uniform data distribution, binarized graphs might omit vital weight information that weighted graphs retain.

$$A_{i,j} = \begin{cases} 1 & \text{if } w_{i,j} \le \epsilon, \\ 0 & \text{otherwise.} \end{cases} \tag{3}$$

$$A = \begin{pmatrix} 0 & w_{0,1} & \cdots & w_{0,N-1} \\ w_{1,0} & 0 & \cdots & w_{1,N-1} \\ \vdots & \vdots & \ddots & \vdots \\ w_{N-1,0} & w_{N-1,1} & \cdots & 0 \end{pmatrix} \tag{4}$$

We regard the type of adjacency matrix as one of the parameters to be optimized. Thus, the sparse coefficient becomes a crucial conditional parameter when the graph type is set to binarized. While there exists no universally accepted guideline for defining the sparsity coefficient for binarized graphs, it is generally observed that

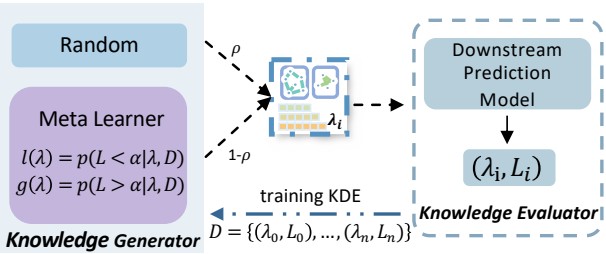

**Figure 3: Meta Learner Overview.**

binarized graphs with excessive connections may inadvertently introduce noise and irrelevant correlations. Consequently, the sparsity coefficients for binarized graphs in our search space are intentionally set to be relatively low.

### 4.2 Knowledge Generator for Sampling

To achieve robust performance, it is crucial to search for the optimal configuration in a huge search space. Our knowledge generator employs two sampling strategies. Firstly, the meta-learner extracts transferable meta-knowledge about optimal ST knowledge configurations and predicts the next likely best configuration. Concurrently, a random sampling strategy with a probability of $\rho$ ensures comprehensive global exploration. This balanced approach integrates targeted learning from the meta-learner and global exploration through random sampling, effectively balancing exploration and exploitation while safeguarding against potential inaccuracies in the meta-learner's predictions [4].

Specifically, meta learner is trained to estimate the mapping from ST knowledge configuration $\lambda$ to the expected performance

$$\mathcal{L}(\lambda, \mathbf{Y}_{t+1}, \mathbf{Y}'_{t+1}). \tag{5}$$

Initially, random sampling is employed to obtain configurations and their corresponding evaluated results from knowledge evaluator, forming the observed data points $D = (\lambda_0, \mathcal{L}_0), \cdots, (\lambda_i, \mathcal{L}_i)$. We then use a multidimensional Kernel Density Estimator (KDE) [4], implemented with statsmodels [25], to model the densities

$$l(\lambda) = p(\mathcal{L} < \alpha | \lambda, D),$$
$$g(\lambda) = p(\mathcal{L} > \alpha | \lambda, D) \tag{6}$$

based on $D$ over the ST knowledge space. In order to train useful KDEs, we need sufficient historical observation, with a minimum number $N_{min}$. After initializing with $N_{min} + 2$ random configurations, we use the KDE to sample configurations with a certain probability $1$-$\rho$. Specifically, we select

$$N_l = \max(N_{min}, q \cdot |D|),$$
$$N_g = \max(N_{min}, |D| - N_l) \tag{7}$$

the best and worst configurations, respectively, to model the two densities, where $q$ is a percentile that trades off the exploration and exploitation, and $|D|$ is the number of observations. This ensures both models have sufficient data points and minimal overlap when observations are limited. To encourage more exploration around

the promising configurations and improve convergence, we consider a KDE model $l'(\lambda)$ similar to $l(\lambda)$, with one major difference that all bandwidths, estimated using Scott's rule of thumb, are multiplied by a factor $b_w$ [4]. Then we sample $N_s$ points from $l'(\lambda)$ and return the sample with the highest ratio $l(\lambda)/g(\lambda)$ as the next candidate configuration. We summarize the above knowledge sampling procedure in Algorithm 2.

---

**Algorithm 2** Knowledge Generation

---

**Require:** observations $D$, $\rho$, $q$, $N_s$, $N_{min}$, $b_w$
**Ensure:** next configuration to evaluate
1: **if** $rand() < \rho$ **or** $|D| < N_{min} + 2\}$ **then**
2:     **return** random configuration
3: **end if**
4: fit the meta learner according to Equation 6 and 7
5: sampling $N_s$ points according to $l'(\boldsymbol{x})$
6: **return** configurtion with highest ratio $l(\boldsymbol{x})/g(\boldsymbol{x})$

---

### 4.3 Knowledge Evaluator for Optimization

The knowledge evaluator executes SUCCESSIVE HALVING algorithm based on the given number of configurations and resources $(n_s, r_s)$. Specifically, we define a function $\mathcal{L}(\mathcal{M}(\lambda), r, \mathbb{D}_{val})$, where $\mathcal{M}$ is the downstream prediction model (e.g., the optional STGNN), $\lambda$ is the knowledge configuration to be evaluated, and the function returns the validation loss after training the configuration for the allocated budget $r$. We also define a function $topk(\Lambda, loss, k)$ to return the top $k$ configurations with the lowest loss (best performance) from the set $\Lambda$, where $k = n/\eta$. Overall, the evaluator uses the $\mathcal{L}$ function to evaluate each configuration and then uses the $topk$ function to eliminate poorly performing configurations. This process leaves $n_{s,i} = n_s \cdot \eta^{-i}$ configurations and increases the allocated budget to $r_{s,i} = r_s \cdot \eta^i$ for each configuration. The next round of evaluation is performed until all the budget is exhausted, where $i$ is the current number of cycles. Overall, knowledge evaluator has a computational complexity of $O(nlogn)$. In our work, the budget is set to the maximum epochs required for training.

## 5 Experiments

In this section, we conduct extensive experiments on real-world datasets to verify the effectiveness of STKOpt and answer the following key research questions.

- **RQ1**: Does STKOpt improve the prediction performance of downstream baselines on different datasets?
- **RQ2**: How effective is the knowledge search space and optimization algorithm?
- **RQ3**: Is there any connection between the ST knowledge learned by STKOpt and the data distribution?
- **RQ4**: How do the hyperparameters of STKOpt influence the ultimate performance?
- **RQ5**: How does STKOpt relate to Neural Architecture Search?

### 5.1 Experimental Settings

*5.1.1 Datasets.* Following previous works [3, 17, 32], we use five public traffic datasets for performance evaluation: METR-LA, PEMS-BAY, BIKE-NYC, TAXI-NYC, LargeST-SD [3, 17, 21, 32], whose

**Table 2: Overall performance comparison on different datasets.**

| Model | Dataset | PEMS-BAY | METR-LA | TAXI-NYC | BIKE-NYC | LargeST-SD |
|---|---|---|---|---|---|---|
| | Metrics | RMSE/MAE | RMSE/MAE | RMSE/MAE | RMSE/MAE | RMSE/MAE |
| DCRNN | | 3.96/1.99 | 11.10/5.97 | 4.42/2.30 | 4.79/3.01 | 34.21/22.56 |
| w/ STKOpt | | **3.81/1.82** | **9.97/5.84** | **4.18/2.18** | **4.51/2.84** | **30.23/20.18** |
| STGCN | | 3.52/1.73 | 11.14/5.52 | 4.56/1.62 | 3.72/2.38 | 27.76/20.21 |
| w/ STKOpt | | **3.39/1.61** | **10.01/5.41** | **4.18/1.54** | **3.61/2.21** | **25.32/19.24** |
| STMeta | | 3.35/1.59 | 10.12/5.30 | 3.08/1.53 | 3.48/2.30 | 23.27/14.11 |
| w/ STKOpt | | **3.31/1.57** | **9.72/4.96** | **2.73/1.40** | **3.42/2.27** | **21.86/13.42** |
| GWN | | 3.60/1.62 | 9.87/4.62 | 2.92/1.35 | 3.55/2.64 | 21.23/12.21 |
| w/ STKOpt | | **3.51/1.53** | **9.70/4.51** | **2.74/1.19** | **3.37/2.51** | **20.19/11.54** |
| D$^2$STGNN | | 3.41/1.49 | 9.86/4.26 | 2.36/1.05 | 3.55/2.27 | 20.72/11.81 |
| w/ STKOpt | | **3.32/1.47** | **9.73/4.54** | **2.01/0.63** | **3.37/2.14** | **20.15/11.21** |
| DGCRN | | 3.32/1.51 | 10.27/4.73 | 2.77/1.27 | 4.38/2.69 | 21.51/13.08 |
| w/ STKOpt | | **3.28/1.50** | **9.66/4.61** | **2.58/1.13** | **3.59/2.29** | **20.37/12.28** |
| STWave | | 3.54/1.60 | 9.95/4.71 | 2.80/1.29 | 4.09/2.58 | 24.75/15.40 |
| w/ STKOpt | | **3.44/1.54** | **9.65/4.57** | **2.56/1.17** | **3.86/2.37** | **23.27/14.29** |

records include traffic speed, taxi order, bike order and traffic flow. We aggregate the sequence into 60-minute windows to study the prediction in a more extended future time period. The granularity of aggregation affects the results, which may lead to some differences between our reported results and those in the original paper. Table 3 summarizes the statistics of the datasets. More details about the datasets is given in Appendix A.1.

**Table 3: Dataset statistics.**

| Dataset | # Timestamps | # Nodes | Time Granularity |
|---|---|---|---|
| PEMS-BAY | 52,128 | 325 | 5 minutes |
| METR-LA | 34,272 | 207 | 5 minutes |
| BIKE-NYC | 446,976 | 820 | 5 minutes |
| TAXI-NYC | 779,904 | 263 | 5 minutes |
| LargeST-SD | 525,888 | 716 | 5 minutes |

*5.1.2 Baselines.* We consider the following baselines, including typical predefined graph-based models (DCRNN [17], STGCN [40], STMeta [32]) and several adaptive learning-based models (GWN [37], D$^2$STGNN [26], DGCRN [13], STWave [5]). We compare the original performance and enhanced performance(w/ STKOpt) of different baselines. More details can be found in Appendix A.2.

*5.1.3 Metrcis.* We leverage Mean Absolute Error (MAE) and Root Mean Squared Error (RMSE) to evaluate the performance of models.

$$RMSE = \sqrt{\frac{1}{N}\sum_{i=1}^{N}(\hat{y}_i - y_i)^2}, \qquad (8)$$

$$MAE = \frac{1}{N}\sum_{i=1}^{N}|\hat{y}_i - y_i|, \qquad (9)$$

where $N$ is the number of nodes, $\hat{y}_i$ is the predicted value, and $y_i$ is the ground truth.

*5.1.4 Implementation Setup.* We adopt two widely used metrics, Root Mean Squared Error (RMSE) and Mean Absolute Error (MAE) [1, 23] to evaluate the performance of models. For optimization, we use RMSE as loss function in knowledge evaluator. Other implementation details are given in Appendix A.3.

## 5.2 Overall Prediction Performance (RQ1)

In this section, we evaluate the original and enhanced performance (w/ STKOpt) of different baselines on five datasets, and the results are shown in the Table 2. We observe that STKOpt improves different types of methods, including predefined graph-based models and models based on adaptive graph learning. This shows that finding optimal ST knowledge is necessary for different model architectures, and this improvement does not partialize to a certain category of baselines, which proves the generalization ability of STKOpt.

Another observation is that the improvements achieved by STKOpt on certain baselines are relatively limited for specific datasets. For instance, the RMSE of GWN improved by only 0.06 on the METR-LA dataset, and the RMSE of STMeta improved by just 0.04 on the PEMS-BAY dataset. A possible explanation for these findings is that the ST knowledge originally employed by these baselines already represents the data distribution well. Specifically, STMeta is designed to combine human knowledge by manually selecting the optimal ST knowledge to achieve good results. Since human efforts are finite, the initial knowledge is likely suboptimal for some dataset, STKOpt can provide significant improvements (e.g., a 0.35 RMSE improvement on the TAXI-NYC dataset) in this case. GWN combines predefined graphs and data-driven graphs to capture hidden spatial dependencies. For high-quality datasets with complete and low-noise data, the data-driven approach can naturally

identify excellent spatial knowledge, making STKOpt's impact less pronounced. However, for datasets where the data-driven methods struggle, STKOpt can effectively bridge this gap.

## 5.3 Ablation Study (RQ2)

In this subsection, we conduct ablation studies for STKOpt. STMeta is utilized as the downstream approach, and two datasets PEMS-BAY and METR-LA are employed for evaluation.

*5.3.1 Effectiveness of search space.* 1) **w/o TK**, which removes temporal knowledge (TK) from search space; 2) **w/o SK**, which removes spatial knowledge (SK) from the ST knowledge search space. As shown in Figure 4, both SK and TK can enhance the forecasting performance in ST prediction problems. In particular, compared to SK, TK has a more significant impact on the final result, as accurately capturing temporal correlation plays a pivotal role in ST prediction problems. This also explains why, in certain scenarios, time series models or machine learning algorithms solely based on temporal knowledge can achieve favorable result.

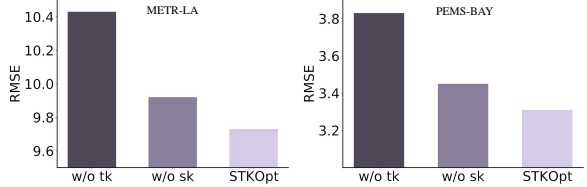

**Figure 4: Ablation Studies on ST knowledge search space.**

*5.3.2 Effectiveness of optimization algorithm.* 1) **STKOpt-R**, which randomly samples knowledge from our search space and evaluate the configuration after complete training; 2) **STKOpt-T**, which trains Tree Parzen Estimator to determine knowledge based on previous observations.

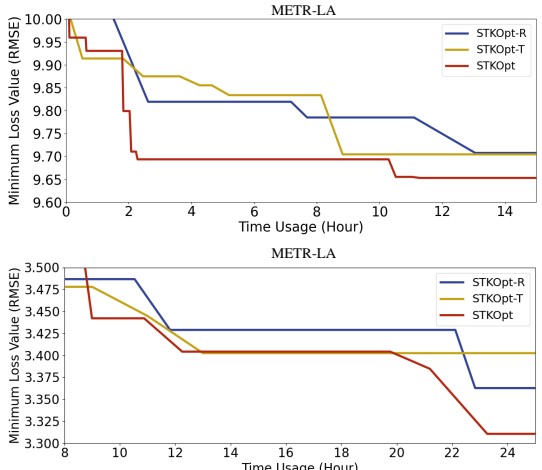

**Figure 5: Ablation Studies on optimization algorithm.**

Figure 5 illustrates the temporal progression of the minimum loss curve. Initially, STKOpt extensively explores the search space

and employs a meta-learner to discover promising configurations. This enables STKOpt to learn from previous results and effectively exploit these configurations, explaining its superior performance compared to STKOpt-R. While STKOpt-T possesses similar learning capabilities, the complexity of downstream model makes it very time-consuming to fully evaluate each configuration, resulting in a slower decrease in loss. In contrast, STKOpt utilizes a multi-fidelity optimization strategy to expedite configuration evaluation and avoid wasting time on configurations with no potential. This strategy begins by evaluating numerous configurations with a smaller budget and then employs successive halving to eliminate underperforming configurations, thereby focusing resources on those with potential. Consequently, although STKOpt may require more initial time to screen potential configurations, it ultimately achieves lower losses more rapidly than STKOpt-T.

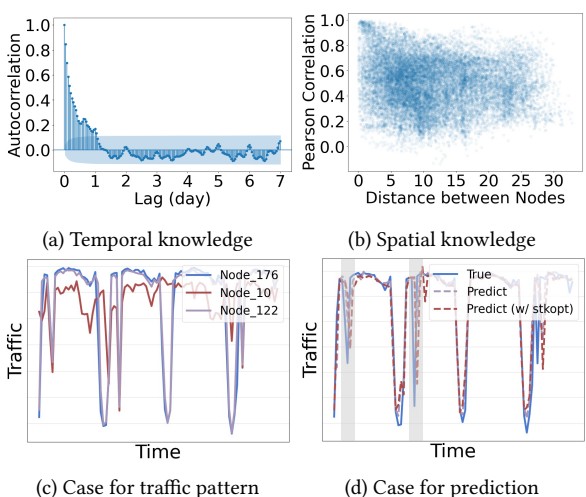

(a) Temporal knowledge     (b) Spatial knowledge

(c) Case for traffic pattern     (d) Case for prediction

**Figure 6: Case study for STKOpt on METR-LA dataset. (a) and (b) present the statistics of the dataset; (c) illustrates that traffic patterns are not always related to distance, and (d) shows the improvement of prediction by the STKOpt framework.**

## 5.4 Investigation on Optimization Effect (RQ3)

We further investigate the relationship between the optimal ST knowledge searched by STKOpt and the data distribution. First, we study the data distribution of METR-LA datasets. Specifically, we use Autocorrelation Function (ACF) of node-wise traffic data to measure the periodicity of traffic time series. As shown in Figure 6(a), contrary to the traditional view that periodicity is inevitable, the periodicity in the METR-LA dataset is not pronounced. Additionally, Figure 6(b) illustrates the relationship between distance between node pairs and their traffic correlation. Conventional wisdom suggests that geographically closer nodes would exhibit stronger traffic correlations. However, the METR-LA dataset lacks a discernible pattern in this regard, displaying a relatively uniform distribution of correlations. This leads us to posit that relying on geographical distance for constructing a spatial knowledge graph may not guarantee enhanced prediction performance.

We analyze the traffic patterns of node 176 and its nearest neighbors (node 10), as well as the node with the strongest correlation (node 122), as illustrated in Figure 6(c). The results reveal that spatial proximity does not necessarily imply similar traffic patterns among nodes, validating our hypothesis. Secondly, we analyze the optimal ST knowledge indentified by STKOpt, i.e., $\{l_c = 8, l_p = 1, l_t = 4, \#graph = 1, \mathcal{G} = correlation, \mathcal{G}_{type} = weighted\}$. The optimal knowledge does not heavily rely on periodic information and utilizes a weighted spatial knowledge graph to preserve weight information, accurately capturing the correlation strength between node pairs and enabling more refined modeling of their positional relationships. Figure 6(d) demonstrates that employing optimal knowledge can effectively improve prediction results, underscoring STKOpt's capability to discover optimal ST knowledge tailored to the unique distributions of diverse datasets.

## 5.5 Hyperparameter Sensitivity Analysis (RQ4)

In this subsection, we evaluate the impact of key hyperparameters in STKOpt, including the size of one unit of budget $b$ (the number of iterations in this paper), and the proportion $\eta$ of configuration discarded in each round of iteration. We conduct experiments on PEMS-BAY dataset and choose STMeta as the downstream model. For each hyperparameter, we show how the prediction results vary with it by fixing other hyperparameters. To be specific, we select the value of $b$ among $\{20, 30, 40\}$ and the value of $\eta$ among $\{3, 4\}$.

Figure 7(a) illustrates how prediction results vary with $b$. As $b$ increases, the duration required to evaluate for each ST knowledge configuration inevitably extends, leading to more time needed to reduce the loss. However, minimizing unit of budget is not always advantageous. A smaller unit of budget implies a reduced time allocation for evaluating configurations, which may bias the evaluation results and miss promising configurations. Furthermore, figure 7(b) shows the impact of $\eta$ values on the results. Tuning $\eta$ will change the number of brackets and consequently the number of different trade-offs mentioned in Section 4.3. Besides, larger $\eta$ also implies that more ST knowledge configurations are discarded in each iteration. Hence, with a fixed total budget B, a larger $\eta$ reduces the number of configurations explored but allocates more resources to each selected configuration.

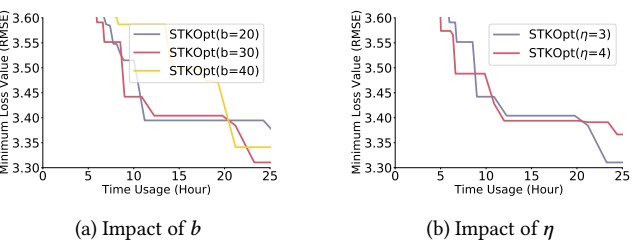

(a) Impact of $b$  (b) Impact of $\eta$

Figure 7: Studies on hyperparameters.

## 5.6 Knowledge Optimization v.s. Architecture Search (RQ5)

We further explore the relationship between knowledge optimization and architecture search. As shown in Figure 8, we compare the running time and memory consumption, respectively. The results show that running STKOpt requires additional time, and the amount of time is closely related to the complexity of the downstream model it is combined with. Additionally, compared to AutoSTG [23], a representative model based on NAS methods, knowledge optimization can elevate an initially average downstream model (e.g., DGCRN) to a competitive level in a shorter time. In terms of memory consumption, running STKOpt essentially does not bring additional memory overhead since it primarily involves fine-tuning the input to the model. In contrast, AutoSTG consumes a large amount of memory, and as the search space becomes more complex and expanded, the memory usage continues to grow because it needs to treat the entire search space as a supernet. Overall, when time and memory budgets are limited, selecting downstream models with relatively simple architectures for knowledge optimization can yield competitive results.

It is worth emphasizing that knowledge optimization and architecture search are not in a competing relationship. The NAS-based method can be viewed as a specific predictive model serving as the downstream model for STKOpt. To verify this hypothesis, we combined STKOpt with AutoSTG. We first searched for the optimal architecture and subsequently identified the optimal ST knowledge to achieve best performance. The results demonstrate that STKOpt can further enhance the performance of AutoSTG, reducing the original RMSE from 10.12 to 9.65. More experimental results can be found in Appendix C.1. In the future, more elegant combination methods remain to be explored.

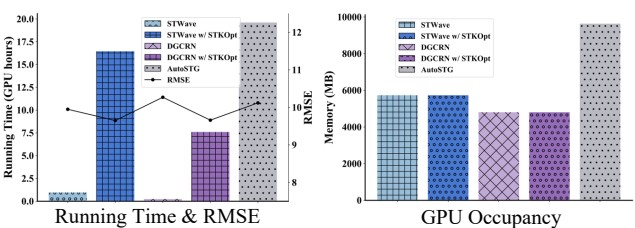

Figure 8: Time consumption and GPU occupancy.

## 6 Conclusion

In this paper, we propose an automated framework tailored for spatio-temporal traffic prediction tasks, entitled STKOpt. It captures the complex ST dependencies by discovering the optimal ST knowledge within our knowledge search space defined by various parameters. To efficiently explore the search space, we train a meta-learner capable of predicting promising knowledge configurations. Additionally, we adopt a bandit-based multi-fidelity algorithm for knowledge optimization to address the constraints of limited time and memory budget. We conduct extensive experiments on five real-world datasets, demonstrating our STKOpt can enhance the performance of downstream prediction models across different tasks by finding optimal ST knowledge.

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

## A Details of the Experiment Setup

### A.1 Datasets

We use five public traffic datasets for performance evaluation.

- **METR-LA**: This dataset is a traffic speed time series dataset recorded by sensors at 207 different locations on highways in Los Angeles County, USA.
- **PEMS-BAY**: This dataset is a traffic speed time series dataset recorded by sensors at 325 different locations and collected by the California Transportation Agencies Performance Measurement System.
- **BIKE-NYC**: This dataset is a bike flow dataset collected from New York City and covers a three month time period.
- **TAXI-NYC**: This dataset is a ride-hailing dataset collected from New York City. The dataset is collected in the community area and covers a three month time period.
- **LargeST-SD**: This dataset is a traffic flow dataset recorded by sensors at 716 different locations on highways in San Diego county, USA.

### A.2 Baseline Details

- **DCRNN** [17]. This model combines diffusion GCN and GRU for traffic prediction.
- **STGCN** [40]. It employs a Chebysev GCN and a gated 1D convolution to build spatio-temporal model.
- **STMeta** [32]. It flexibly integrates multiple spatio-temporal knowledge and can generally work well on various scenarios.
- **GWN** [37]. It designs a data-driven graph convolution method for adaptively learning spatial knowledge and captures spatial-temporal dependencies by combining graph convolution with dilated casual convolution.
- **D$^2$STGNN** [26]. This model proposes a novel decoupled spatio-temporal framework and then design the diffusion and inherent model as well as the dynamic graph learning model.
- **DGCRN** [13]. This model proves that dynamic graph can effectively cooperate with pre-defined graph while improving the prediction performance.
- **STWave** [5]. This model proposes a disentangle-fusion framework and incorporate a query sampling strategy and graph wavelet-based graph positional encoding to model the dynamic spatial correlations.

### A.3 Implementation Details

*A.3.1 Knowledge Search Space Setup.* The number and types of candidate graphs in the search space are adjustable based on different data conditions. For example, not all datasets include geographic attributes such as points of interest (POIs). Therefore, in this paper,

we select two types of graphs that can be constructed on most datasets: distance graphs and correlation graphs.

- We construct weighted distance graph using thresholded Gaussian kernel as follows:

$$W_{ij} = \begin{cases} \exp\left(-\dfrac{\text{dist}(v_i,v_j)^2}{\sigma^2}\right), & \text{if } \text{dist}(v_i, v_j) \leq \epsilon \\ 0, & \text{otherwise} \end{cases}$$

where $\text{dist}(v_i,v_j)$ represents the distance between nodes $v_i$ and $v_j$; $\sigma$ is the standard deviation of distances and $\epsilon$ is the sparsity coefficient to be optimized to control the sparsity of graph.

- We compute the correlations between every two nodes to construct correlation graph using Pearson correlation coefficient.

$$W_{ij} = |\frac{\sum_{i=1}^{n}(X_i - \overline{X})(Y_i - \overline{Y})}{\sqrt{\sum_{i=1}^{n}(X_i - \overline{X})^2}\sqrt{\sum_{i=1}^{n}(Y_i - \overline{Y})^2}}|$$

For the correlation graph, in addition to the weighted adjacency matrix, we also consider its binarized matrix. We use a sparsity coefficient $\epsilon$ to control the sparsity of graph. When $\epsilon$ is set to 0.1, only the top 10% most relevant edges are retained.

*A.3.2 Meta Learner Setup.* We set the minimum number of sample points $N_{min}$ required to build a meta-learner as $d + 1$, where d is the number of parameters in our proposed search space. The proportion of best configurations $q$ in observations $D$ is set to 15%. To improve EI, the bandwidth widen factor $b_w$ is set to 1e-3 and the number of drawn samples $N_s$ is set to 64. For sampling the next configuration, we set the proportion of random sampling $\rho$ as 1/3.

*A.3.3 Optimization Algorithm Setup.* We employ distinct optimization parameters tailored to different downstream models. Our budget adjustments are based on the resources utilized in the original paper, i.e., the number of epochs. For DCRNN, STGCN, STMeta, GWN, the maximum budget $B$ is 100, with one unit of budget $b$ being 30, and the proportion of configuration discarded in each round of iteration $\eta$ being 3. For DGCRN, $b$ is 1, while the other parameters remain the same as above. For D$^2$STGNN, $B$ is 81, $b$ is 1 and $\eta$ remains the same. For STWave, $b$ is 6, with the other parameters consistent with the aforementioned models.

*A.3.4 Downstream Model Setup.* For different models, we adopt the optimization algorithms and learning rates as specified in the original papers. For different datasets, we select varying time spans: we adopt the entire time span for the PEMS-BAY and METR-LA datasets, the first three months for the BIKE-NYC and TAXI-NYC datasets, and the year 2019 for the LargeST-SD dataset. The batch size is set to 64 for the METR-LA, PEMS-BAY, and TAXI-NYC datasets, 32 for the BIKE-NYC dataset, and 16 for the LargeST-SD dataset.

## B Details of Optimization Framework

In theory, the maximum speedup offered by STKOpt compared to random search is $\frac{B}{\lceil \log_\eta(B) \rceil + 1}$. However, in practice, the actual speedup is influenced by two main factors: 1) The relationship between training time and allocated resources: When training time scales superlinearly with respect to the resource, STKOpt can achieve higher speedups; 2) Additional overhead associated

**Table 4: Overall performance comparison on different datasets.**

| Model | PEMS-BAY | | METR-LA | | TAXI-NYC | |
|---|---|---|---|---|---|---|
| | RMSE | MAE | RMSE | MAE | RMSE | MAE |
| DCRNN | 3.96±0.02 | 1.99±0.01 | 11.10±0.04 | 5.97±0.02 | 4.42±0.01 | 2.30±0.01 |
| w/ STKOpt | **3.81±0.02** | **1.82±0.01** | **9.97±0.04** | **5.84±0.02** | **4.18±0.04** | **2.18±0.03** |
| STMeta | 3.35±0.01 | 1.59±0.00 | 10.12±0.08 | 5.30±0.06 | 3.08±0.04 | 1.53±0.02 |
| w/ STKOpt | **3.31±0.02** | **1.57±0.01** | **9.72±0.06** | **4.96±0.04** | **2.73±0.06** | **1.40±0.04** |
| DGCRN | 3.32±0.02 | 1.51±0.01 | 10.27±0.08 | 4.73±0.04 | 2.77±0.09 | 1.27±0.06 |
| w/ STKOpt | **3.28±0.02** | **1.50±0.01** | **9.66±0.08** | **4.61±0.05** | **2.58±0.06** | **1.13±0.04** |
| STWave | 3.54±0.02 | 1.60±0.01 | 9.95±0.02 | 4.71±0.02 | 2.80±0.01 | 1.29±0.01 |
| w/ STKOpt | **3.44±0.02** | **1.54±0.02** | **9.65±0.06** | **4.57±0.04** | **2.56±0.04** | **1.17±0.03** |

with training: Beyond the time required for training the model, the total evaluation time also includes overhead from model initialization, validation error computation, and the training and sampling time associated with the meta-learner.

## C  Additional Experiments

### C.1  Combination of Knowledge Search and Architecture Search

We compare the experimental results of some baselines with AutoSTG [23], Auto-DSTSGN [10] and AutoCTS [33], as shown in the Table 5. The results show that even though NAS-based methods sometimes outperform the original baselines, most baselines perform better after applying knowledge optimization (w/ STKOpt), proving STKOpt's effectiveness. In addition, we compared time consumption and GPU occupancy of AutoCTS, shown in Figure 9. AutoCTS takes less time than AutoSTG and STWave w/ STKOpt but has the largest RMSE. AutoCTS also requires more memory than the other models.

**Table 5: Performance comparison on different datasets (The best results are marked with '*').**

| Model | Dataset Metrics | PEMS-BAY RMSE/MAE | METR-LA RMSE/MAE | TAXI-NYC RMSE/MAE |
|---|---|---|---|---|
| GWN | | 3.60/1.62 | 9.87/4.62 | 2.92/1.35 |
| w/ STKOpt | | **3.51/1.53** | **9.70/4.51** | **2.74/1.19** |
| DGCRN | | 3.32/1.51 | 10.27/4.73 | 2.77/1.27 |
| w/ STKOpt | | **3.28*/1.50*** | **9.66/4.61** | **2.58/1.13** |
| STWave | | 3.54/1.60 | 9.95/4.71 | 2.80/1.29 |
| w/ STKOpt | | **3.44/1.54** | **9.65*/4.57*** | **2.56/1.17** |
| AutoSTG | | 3.401/1.52 | 10.12/4.68 | 3.57/1.66 |
| AutoCTS | | 5.73/2.75 | 11.01/5.77 | 4.18/1.79 |
| Auto-DSTSGN | | 3.47/1.57 | 10.64/5.03 | 2.43*/1.10* |

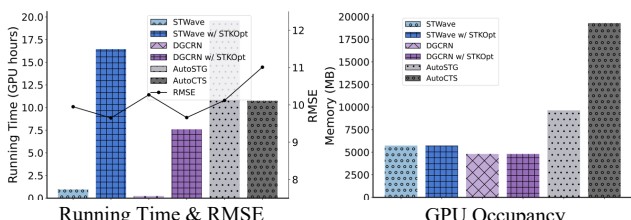

**Figure 9: Time consumption and GPU occupancy.**

### C.2  Necessity of Multiple Sub-Loops

In this section, we analyze the optimization effects of different sub-loops within the STKOpt framework. Each sub-loop represents the evaluation of varying numbers of configurations, using different amounts of resources. The primary objective of employing multiple sub-loops is to address the "$n$ versus $B/n$" dilemma, which pertains to finding the optimal balance between the number of configurations $n$ and the budget allocated per configuration $B/n$. The results, depicted in Figure 10, demonstrate that the optimal prediction outcomes for different datasets occur in distinct sub-loops. This variation underscores the necessity of utilizing multiple sub-loops in STKOpt to achieve the best predictive performance across diverse datasets.

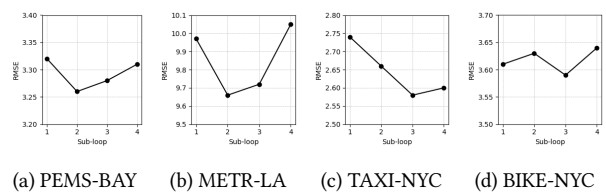

(a) PEMS-BAY    (b) METR-LA    (c) TAXI-NYC    (d) BIKE-NYC

**Figure 10: Necessity analysis of multiple sub-loops.**

### C.3  Replication Study

We train and test each downstream model five times using different random seeds, and present the results in the format of "mean ± standard deviation". The results shown in Table 4 demonstrate that

that STKOpt exhibits strong adaptability to different parameter initialization settings, reliably discovering and providing optimal spatio-temporal knowledge for downstream models.

## D    Limitations and Future Work

The proposed STKOpt has demonstrated its effectiveness in enhancing the prediction performance of downstream models. However, it has two main limitations. First, there is an additional time cost. Knowledge optimization requires extra time, which may limit its applicability in scenarios with strict time constraints. Second, the integration with architecture search methods is suboptimal. Knowledge optimization and architecture search are two orthogonal approaches to address spatio-temporal heterogeneity. Currently, STKOpt combines them sequentially—first conducting architecture search and then applying knowledge optimization. This approach is both time-consuming and fails to fully leverage the strengths of each method. In future research, we aim to develop more efficient spatiotemporal knowledge optimization frameworks and integrate knowledge optimization and architecture search in a more synergistic manner.

