# OpenReview forum: "STKOpt: Automated Spatio-Temporal Knowledge Optimization for Traffic Prediction"
_ACM.org/TheWebConf/2025/Conference — WWW 2025 Poster_

### Official Review · Reviewer_FUhJ · 2024-11-29

**Novelty:** 5
**Technical Quality:** 5

**Review:**

This paper introduces STKOpt, an automated framework for spatio-temporal traffic prediction. STKOpt enhances downstream model performance by discovering optimal spatio-temporal knowledge through a defined search space of parameters that capture scenario-specific correlations. It employs a meta-learner to efficiently explore search space and utilizes a bandit-based multi-fidelity algorithm to optimize knowledge while managing time and memory constraints. Extensive experiments on five real-world datasets demonstrate STKOpt's effectiveness.

Pros:
1. The paper proposes the first framework to automatically discover optimal ST knowledge for spatio-temporal traffic prediction.
2. The paper atomizes spatiotemporal knowledge and designs a knowledge search space composed of parameters, employing a multi-fidelity optimization strategy to solve the constraint of limited resources and utilizing a meta-learner to efficiently explore the search space.
3.	The paper conducts extensive experiments on real-world datasets, and the experimental results demonstrate that STKOpt can enhance the performance of downstream models.

Cons:
1. The description about initializing with $N_{min} + 2$ random configurations is not clear. The author should have more explanation.
2. Although STKopt will improve prediction performance, the spatio-temporal knowledge it captured should have a more intuitive case.
3. How do you determine the optimal number and type of spatial knowledge graphs to include in the model?

**Questions:**

See above.

**Reviewer Confidence:**

2: The reviewer is willing to defend the evaluation, but it is likely that the reviewer did not understand parts of the paper

**Scope:**

3: The work is somewhat relevant to the Web and to the track, and is of narrow interest to a sub-community

---

### Official Review · Reviewer_EVGS · 2024-11-30

**Novelty:** 4
**Technical Quality:** 4

**Review:**

This paper introduces the STKOpt framework, designed to address the challenge of spatiotemporal knowledge optimization in traffic prediction tasks. By focusing on optimizing input knowledge rather than solely refining model architectures, STKOpt provides a novel perspective that complements existing spatiotemporal graph neural networks (STGNNs). The framework employs a combination of techniques, including HYPERBAND for multi-fidelity optimization and meta-learning to leverage transferable knowledge, enabling efficient exploration of a high-dimensional search space. However, the increased computational cost, especially for complex downstream models, may be a factor to consider in resource-constrained or time-sensitive applications.

**Pros:**

1. The paper shifts the focus to optimizing spatiotemporal knowledge as input, addressing a critical but often overlooked aspect in existing studies on spatiotemporal graph neural networks.
2. Comprehensive experiments on diverse datasets and benchmarks demonstrate the robustness and generalizability of the proposed framework.
3. The integration of multi-fidelity optimization and meta-learning reduces computational overhead compared to exhaustive search methods, balancing efficiency and performance.

**Cons:**

1. Although ablation studies are included, the specific contributions of key components, such as the meta-learning module, are not analyzed in depth, limiting the interpretability of the results.
2. Despite efforts to improve computational efficiency, the framework shows a noticeable increase in training time, particularly for complex downstream models, which could hinder its applicability in time-sensitive scenarios or tasks requiring rapid deployment.
3. The author should consider more representative baselines presented in recent (last) years.

**Questions:**

1. Can you provide theoretical justification and specific reasoning for the selection of key parameters, such as the sparsity coefficient for spatial graphs?
2. How does the framework handle scenarios with high noise or sparse data? Would additional preprocessing steps alleviate these challenges?

**Reviewer Confidence:**

3: The reviewer is confident but not certain that the evaluation is correct

**Scope:**

3: The work is somewhat relevant to the Web and to the track, and is of narrow interest to a sub-community

---

### Official Review · Reviewer_5Vfd · 2024-11-30

**Novelty:** 4
**Technical Quality:** 5

**Review:**

Summary

This paper proposes an automated spatio-temporal (ST) knowledge optimization module that integrates seamlessly with existing spatio-temporal graph neural networks (STGNNs) to enhance their performance.

Pros

1. The idea of automatically optimizing spatio-temporal knowledge for traffic prediction tasks is innovative.
2.  The paper presents a thorough set of experiments to demonstrate the effectiveness of the proposed method, which is convincing and well-supported.

Cons

1.  STKOpt directly employs the HYPERBAND method for Knowledge Optimization and the BOHB method for Knowledge Generation. However, the innovation of the methods is limited.
2. The symbol definitions in the paper are complex and unclear. For example, in Equation (2), it is uncertain whether "$\lambda^* = \arg\min _{\lambda \in \Lambda} \mathcal{L}(\lambda, \mathbb{D} _ {val})$" should be   "$\lambda^* = \arg\min _{\lambda \in \Lambda} \mathcal{L}(\lambda, \mathbb{D} _ {train})$". Additionally, in Equation (6), the meaning of "$\alpha$" is not properly defined.
3. There are typo errors in lines 335 and 339, such as the terms "atomize" and "atomization."
4. In the "Overall Prediction Performance" section, the experiments only compare STKOpt as an incremental module. It would be beneficial to include comparisons with other AutoML methods for spatiotemporal data, or with state-of-the-art NAS-based methods, to ensure a more fair and comprehensive comparison (e.g., the results presented in Table 5 of the Appendix).

**Questions:**

Q1. Is the optimized spatio-temporal knowledge represented as a static graph structure, or does it evolve over time into a dynamic graph? This distinction should be clarified to improve understanding.

Q2. How do temporal and spatial knowledge align during the optimization process? A more detailed explanation of this mechanism would strengthen the technical contribution.

Q3. The proposed STKOpt focuses on selecting optimal spatio-temporal knowledge as model inputs, while adaptive graph-based methods also aim to optimize graph structures tailored to the ST model. What is the fundamental difference between these two methods? Additionally, the comparison experiment only includes Graph WaveNet (2019) as a baseline for adaptive graph-based methods. It would be more convincing and fair to include comparisons with more recent adaptive graph-based methods.

**Reviewer Confidence:**

3: The reviewer is confident but not certain that the evaluation is correct

**Scope:**

3: The work is somewhat relevant to the Web and to the track, and is of narrow interest to a sub-community

---

### Official Review · Reviewer_wH5w · 2024-12-01

**Novelty:** 5
**Technical Quality:** 4

**Review:**

Pros:
1. This paper presents a general framework, STKOpt, aimed at automating spatio-temporal knowledge optimization for traffic prediction tasks. Extensive experiments have been conducted to verify the effectiveness of the proposed method. The paper addresses the challenge of optimizing inputs for different data scenarios in spatio-temporal traffic prediction. If broadly validated, the proposed framework could have a significant impact on the design of future AutoML systems for urban analytics.
2. The novelty of this work lies in shifting the focus from optimizing model architectures to optimizing spatio-temporal knowledge inputs, which is a less explored area in traffic prediction. Compared to traditional AutoML and NAS methods, STKOpt offers a unique perspective.

Cons:
1. The time consumption brought by the proposed method is substantial compared to the improvements in performance metrics, which represents a limitation of the approach.
2. The paper does not provide enough details on downstream task settings. It is recommended to include a more detailed description of the experimental downstream tasks and also to explore whether the method shows generalizability across various task settings.

**Questions:**

Aggregation and Noise Impact:
In the experimental section, the datasets are aggregated into 60-minute intervals, which likely reduces data noise and makes the prediction task easier. I am curious whether the authors have experimented with the original, unaggregated datasets, and if so, what results were obtained? It would be insightful to know how the model performs without the aggregation step.

Code Availability:
Will the authors be releasing the code for the STKOpt framework in the future?

**Reviewer Confidence:**

3: The reviewer is confident but not certain that the evaluation is correct

**Scope:**

3: The work is somewhat relevant to the Web and to the track, and is of narrow interest to a sub-community

---

### Official Review · Reviewer_MCwn · 2024-12-03

**Novelty:** 6
**Technical Quality:** 6

**Review:**

The paper presents STKOpt, an innovative automated framework for optimizing spatio-temporal knowledge in traffic prediction, addressing a critical gap in the field. It introduces a multi-fidelity optimization strategy and a meta-learner to efficiently explore a large search space, enhancing prediction performance of downstream models. The rigorous methodology and extensive experiments on real-world datasets demonstrate the framework's effectiveness, leading to significant improvements in stability and cost savings. Despite the simplicity of Figure 2 detracting from the visual representation of the framework, the paper's technical soundness, original contributions, and practical implications for smart city applications make it a high-quality work with broad interest to the community.

Strengths:
1. The paper introduces STKOpt, an automated framework for optimizing spatio-temporal knowledge in traffic flow prediction, which is pioneering in its field. The approach of automating the selection of optimal model inputs to enhance prediction performance is innovative.
2. The multi-fidelity optimization strategy and the meta-learner proposed in the paper provide effective technical means for efficiently exploring a large search space under limited resources.
3. The paper validates the effectiveness of the STKOpt framework through extensive experiments on five real-world datasets, which strengthens the persuasiveness of the research findings.
4. The STKOpt framework's ability to enhance the performance of downstream models is of significant practical value for traffic management systems in smart cities and IoT services.

Weaknesses:
1. Figure 2, as the framework diagram, is overly simplistic and fails to adequately illustrate the complexity and detailed workflow of the STKOpt framework, which is a significant hindrance for readers to understand the contributions and methodology of the paper.
2. Although the paper shows performance improvements of various baseline models with STKOpt, it lacks a comprehensive comparison with existing state-of-the-art methods, especially in terms of computational efficiency and model generalization capabilities.
3. The paper does not delve deeply into the limitations of STKOpt, particularly regarding its scalability and computational resource consumption when dealing with large-scale datasets.

**Questions:**

1. Can the authors provide a more detailed framework diagram or additional figures to supplement the explanation of the STKOpt workflow and component interactions?
2. Can the authors discuss the limitations of STKOpt in more detail and propose potential directions for future improvements, particularly in the application to large-scale datasets and real-time prediction scenarios?
3. Can the authors provide more data on the computational resource consumption of STKOpt under different configurations, including memory usage and processing time?

**Reviewer Confidence:**

3: The reviewer is confident but not certain that the evaluation is correct

**Scope:**

4: The work is relevant to the Web and to the track, and is of broad interest to the community